# Bacterial IAA-Delivery into Medicago Root Nodules Triggers a Balanced Stimulation of C and N Metabolism Leading to a Biomass Increase

**DOI:** 10.3390/microorganisms7100403

**Published:** 2019-09-29

**Authors:** Roberto Defez, Anna Andreozzi, Silvia Romano, Gabriella Pocsfalvi, Immacolata Fiume, Roberta Esposito, Claudia Angelini, Carmen Bianco

**Affiliations:** 1Institute of Biosciences and BioResources, via P. Castellino 111, 80131 Naples, Italy; roberto.defez@ibbr.cnr.it (R.D.); anna.andreozzi@ibbr.cnr.it (A.A.); silvia.romano@ibbr.cnr.it (S.R.); gabriella.pocsfalvi@ibbr.cnr.it (G.P.); immacolata.fiume@ibbr.cnr.it (I.F.); 2Institute of Genetics and Biophysics “A.B.T.”, CNR, via P. Castellino 111, 80131 Naples, Italy; roberta.esposito@dbmr.unibe.ch; 3Institute for Applied Mathematics “Mauro Picone” IAC, CNR, via P. Castellino 111, 80131 Naples, Italy; claudia.angelini@cnr.it

**Keywords:** Indole-3-acetic acid, nitrogen-fixation genes, photosynthetic pigments, nitrogen content, metabolic patterns, plant yield

## Abstract

Indole-3-acetic acid (IAA) is the main auxin acting as a phytohormone in many plant developmental processes. The ability to synthesize IAA is widely associated with plant growth-promoting rhizobacteria (PGPR). Several studies have been published on the potential application of PGPR to improve plant growth through the enhancement of their main metabolic processes. In this study, the IAA-overproducing *Ensifer meliloti* strain RD64 and its parental strain 1021 were used to inoculate *Medicago sativa* plants. After verifying that the endogenous biosynthesis of IAA did not lead to genomic changes during the initial phases of the symbiotic process, we analyzed whether the overproduction of bacterial IAA inside root nodules influenced, in a coordinated manner, the activity of the nitrogen-fixing apparatus and the photosynthetic function, which are the two processes playing a key role in legume plant growth and productivity. Higher nitrogen-fixing activity and a greater amount of total nitrogen (N), carbon (C), Rubisco, nitrogen-rich amino acids, soluble sugars, and organic acids were measured for RD64-nodulated plants compared to the plants nodulated by the wild-type strain 1021. Furthermore, the RD64-nodulated plants showed a biomass increase over time, with the highest increment (more than 60%) being reached at six weeks after infection. Our findings show that the RD64-nodulated plants need more substrate derived from photosynthesis to generate the ATP required for their increased nitrogenase activity. This high carbohydrate demand further stimulates the photosynthetic function with the production of molecules that can be used to promote plant growth. We thus speculate that the use of PGPR able to stimulate both C and N metabolism with a balanced C/N ratio represents an efficient strategy to obtain substantial gains in plant productivity.

## 1. Introduction

It is well-established that several agronomically important traits, such as plant height, grain number, and grain filling, are primarily regulated by phytohormones, such as cytokinins, gibberellins, brassinosteroids, and auxins. The main plant auxin, indole-3-acetic acid (IAA), was identified in 1944 [1] and since then, various aspects related to its metabolism, transport, and signaling have been clarified. In the last few years, plant biologists have exploited the information regarding the phytohormone transduction signals to modulate hormone levels and metabolism in plant tissues in order to reduce the damaging effects of external stresses, such as drought, salinity, nutrient deficiency, etc. [2]. Furthermore, several investigations have shown that phytohormones produced by root-associated microbes, including those of the plant growth-promoting rhizobacteria (PGPR), could be important engineered targets for improving plant growth and metabolism [3]. However, due to the complexity of the rhizosphere environment, the multiple plant–soil–microbial (tripartite) interactions, the interplay of hormone balance, and the frequent use of defective mutants rather than strains over-expressing key traits, the observed results have often been ambiguously interpreted. In legume plants, carbon is assimilated during photosynthesis, while nitrogen is provided through the symbiotic fixation of atmospheric nitrogen, by the nitrogenase enzyme in rhizobial bacteroids. In these plants, the nitrogen fixation by bacteroids within root nodules and the assimilation of fixed-N into amino acids are tightly linked to carbon metabolism [4,5,6]. The actively operating symbiotic apparatus, with its high demand for photosynthates, is thought to promote the export of carbohydrates and reserve molecules from leaves, thereby stimulating the photosynthetic function and increasing the plant yield potential [7,8,9,10,11]. It is widely accepted that Rubisco, short for ribulose-1,5-biphosphate carboxylase/oxygenase, is the ultimate rate-limiting step in photosynthetic carbon fixation. It is characterized by a relatively slow catalytic turnover rate, and as a consequence, a large amount of the enzyme is required to sustain adequate photosynthetic rates. Optimizing the Rubisco functionality has considerable implications for the improvement of plant productivity and resource-use efficiency [12,13]. Genetically engineered plants, producing increased levels of Rubisco protein, could potentially improve CO_2_ fixation [9,14]. The photosynthetic pigments produced by the plants, such as chlorophyll a and b and carotenoids, allow them to utilize the most received light, providing photoprotection and stability of the proteins present in the photosystem [15,16,17,18,19,20,21]. We have previously reported that Medicago root nodules hosting RD64, a derivative of the *Ensifer meliloti* 1021 engineered to overproduce the auxin IAA [22,23], showed a more active meristem and increased root nodule zones [24], within which the bacterial IAA was produced even in the oldest tissue of senescent zone IV [25]. Beneficial effects on N-fixation under salt tolerance and low mineral phosphate were also observed for Medicago plants nodulated by the strain RD64 (*Ms*-RD64) compared to 1021-nodulated plants (*Ms*-1021) [26,27]. It is known that plant growth correlates with the net C gain on the whole plant basis and leaf N content and light intensity are the major determinants of the photosynthetic rate. The main objectives of this work were to evaluate whether the synthesis and release of higher IAA levels inside root nodules of RD64-nodulated plants influenced the processes of carbon and nitrogen assimilation over time in a coordinated way and whether the resulting C/N ratio was unbalanced. The results reported show that the expression of nitrogen-fixation genes and the activity of the nitrogenase enzyme were significantly increased in *Ms*-RD64 plants compared to the 1021-nodulated ones. Greater amounts of Rubisco, photosynthetic pigments, nitrogen-rich amino acids, soluble sugars, and organic acids was also measured for these plants. Consistently, the total C and N content increased in a balanced manner, as their relative ratio did not change. As a result, a significant biomass increase was registered for *Ms*-RD64 plants. This work highlights that the use of PGPR, triggering a balanced stimulation of C and N metabolism, could be an efficient strategy for improving plant growth.

## 2. Materials and Methods

### 2.1. Bacterial Strains, Growth Conditions, and Plasmids

The bacterial strains used in this study are the *Ensifer meliloti* 1021 [28] and the *E. meliloti* RD64. RD64 is an IAA-overproducing strain harboring the pMB393 plasmid containing the p-*iaaMtms2* construct in which the coding region of the *iaaM* gene of *P. savastanoi* is positioned downstream from an 85 bp promoter sequence, and the *tms2* coding region of *A. tumefaciens* is placed at the 3′ end of the *iaaM* gene [22,23]. Bacterial strains (*E. meliloti* 1021 and RD64) used for the inoculation of Medicago plants were aerobically grown at 30 °C in TYR medium (5 g/L tryptone, 3 g/L yeast extract, 6 mM CaCl_2_) [27]. Streptomycin (200 mg/L) and spectinomycin (200 mg/L) were included as required. The modified strain RD64 releases about 90% more IAA (56 µM) into the bacterial culture supernatant than the parental strain 1021 (0.60 µM) [25].

### 2.2. Nucleic Acid Preparation from Bacteroids

Six weeks after bacterial inoculation, alfalfa roots were subjected to surface sterilization at room temperature as follows: (1) 1 min in 0.15% (*v*/*v*) sodium hypochlorite (NaClO); (2) washing stages using water to remove the NaClO adhered on the surface; (3) 1 min in EtOH 70% (*v*/*v*); (4) several washing stages with water. Nodules (3 g) were then collected, put into liquid nitrogen, and ground using a pre-chilled pestle in 10 mL of extraction buffer (40 mM MOPS buffer pH 7.0, 300 mM sucrose, 2 mM MgSO4, 20 mM KOH, and 3% (*w*/*v*) polyvinylpirrolidone (PVP). The resulting homogenate was passed through three layers of gauze, and the filtrate was centrifuged at 500× *g* (5 min; 4 °C) to remove large particles of plant cell debris. Then, the supernatant was centrifuged at 4600× *g* (15 min; 4 °C) to collect bacteroids. Bacteroids were washed twice with 5 mL of extraction buffer, centrifuged at 4000× *g* (10 min; 4 °C), and examined by light microscopy to ensure that they were substantially free from contaminating plant material. After microscopy checking, bacteroids were then used for nucleic acid preparation. Genomic DNA was extracted by using the Wizard Genomic DNA Purification kit (Promega, Madison, WI, USA), according to the manufacturer’s instructions. After purification and quality checking by agarose gel electrophoresis, the DNA concentration was determined by absorbance at 260 nm and the nucleic acids were stored at −20 °C until further use.

### 2.3. Sequencing

Sequencing libraries were constructed using the TruSeq DNA Sample Prep Kit (Illumina, San Diego, CA, USA), according to manufacturer’s instructions. DNA-seq libraries were sequenced paired-end 100 bp at a 3-plex level of multiplexing on a HiSeq2000 Illumina sequencer at IGA Technology Services (Udine, Italy). Raw images were processed using *Illumina Pipeline* version 1.8.2 (Illumina, San Diego, CA, USA).

### 2.4. Mapping of Sequenced Reads and Data Analysis

Raw sequence data from high-throughput sequencing pipelines were analyzed using FastQC version 0.10.1 (http://www.bioinformatics.bbsrc.ac.uk/projects/fastqc), which is a tool employed for quality control checks. Raw sequence reads of the 1021 and RD64 bacteroids were trimmed of 25 bp, according to their quality score, using NGS QC Toolkit version 2.3.1 [29]. The reference genome of *S. melitoti* 1021 was downloaded from NCBI (ftp://ftp.ncbi.nlm.nih.gov/genomes/Bacteria/Sinorhizobium_meliloti_1021_uid57603/) and used to map the reads. Sequenced reads, for each sample, were aligned to the reference genome using Bowtie2 version 2.0.6 [30], an ultrafast and memory-efficient tool for aligning sequencing reads to long reference sequences. Only unique and concordant mapping reads were used for subsequent analysis. PCR duplicate reads were removed using Picard tools version1.93 (http://broadinstitute.github.io/picard/). Subsequently, three different tools were used for single nucleotide variants (SNVs) and small insertion and deletion (INDEL) calling: VarScan version 2.3.5 [31], FreeBayes version 0.9.9.2 [32], and SAMtools version 0.1.19 [33]. Details on the parameters used are reported as Appendix A. The Integrated Genomics Viewer [34] was used to visually check the quality of the alignments on the S. *meliloti* 1021 reference genome. In order to filter the resulting call set, variants with clusters of at least three SNVs that were within a window of 35 bases and variants with a Phred Quality <30 were filtered out. In order to assess the effects of putative variants on genes, transcripts, and the protein sequence, SNVs and INDELs were annotated using the Variant Effect Predictor tool of Ensembl (http://bacteria.ensembl.org). The raw sequencing data were deposited in the NCBI Short Read Archive database under the accession number SRX800905.

### 2.5. SNV Validation by Sanger Sequencing

Next-generation sequencing results were validated by Sanger sequencing. Sanger sequencing was carried out on a random selection of variants identified through our sequencing pipeline in both the 1021 and RD64 bacteroids. Primer pairs flanking each SNV were designed using Primer3 software (http://primer3.sourceforge.net/). Template DNA (50 ng), purified as described in the previous section, was then used for amplification with the *Euro*Taq PCR protocol. Following amplification, PCR products were visualized on 1.5% (*w*/*v*) agarose gels, and those showing a single clean band in the proper size range were selected for further processing. PCR products were then subjected to Sanger sequencing. Specific primer pairs for selected genes, designed using Primer3 software, were reported in Appendix A.

### 2.6. qRT-PCR Analysis

To isolate RNA from plant nodules, frozen tissues were homogenized in 500 µL QIAzol lysis reagent (Qiagen, Hilden, Germany) and centrifuged at 12,000× *g* for 1 min at 4 °C. Chloroform (100 µL) was then added to the clear supernatant, mixed well, and centrifuged at 12,000× *g* for 15 min. The transparent upper phase was mixed with an equal volume (300 µL) of 70% ethanol, transferred to an RNeasy Mini spin column (RNeasy minikit, Qiagen, Hilden, Germany), and centrifuged at 8000× *g* for 15 s. After the addition of Buffer RW1 (350 µL) to the column, the manufacturer’s instruction was followed. Residual DNA present in the RNA preparations was removed by incubating it at 37 °C with 2 U of TURBO DNase and 20 U of RNAse inhibitor (TURBO DNA-*free* I Kit, Applied Biosystems, Waltham, MA, USA) for 30 min. After purification and quality checking by agarose gel electrophoresis, the RNA concentration was determined by absorbance at 260 nm and the RNA was stored at −20 °C until further use. First-strand cDNA was synthesized from 1 µg of total RNA by incubating it with 100 U of MMLV-RT, 4 U of RNAse inhibitor and 5 µM Random Decamers for 1 h at 44 °C (RETROscript kit, Applied Biosystem, Waltham, MA, USA). qRT-PCR was carried out with the iQ SYBR green Supermix (Bio-Rad, Hercules, CA, USA) and the iCycler iQ (Bio-Rad, Hercules, CA, USA). The thermocycling conditions were as follows: 15 min at 95 °C, 35 cycles of denaturation at 95 °C for 20 s and annealing at specific temperatures for 20 s, and extension for 35 s at 72 °C. Specific primer pairs used for this analysis were designed using Primer3 software (version 0.4.0, http://primer3.sourceforge.net/) and are reported in the Appendix A. Primers for *Actin* [26] and *rpoB* [35] were included in all the qRT-PCR analyses for the purpose of data normalization. During the reactions, the fluorescence signal due to SYBR Green intercalation was monitored to quantify the double-stranded DNA product formed in each PCR cycle. The results were recorded as relative gene expression changes after normalizing for *Actin* and *rpoB* gene expression and were computed using the comparative CT method (2^–ΔΔC*T*^), as previously described in Livak and Schmittgen [36]. For free-living cells, the 2^–ΔΔC*T*^ value was >1 for genes more highly expressed in IAA-treated or Novobiocin-treated 1021 cells and <1 for genes more highly expressed in untreated 1021 cells. For nodulated plants, the 2^–ΔΔC*T*^ value was >1 for genes more highly expressed in *Ms*-RD64 root nodules and <1 for genes more highly expressed in *Ms*-1021 wild-type root nodules. qRT-PCR data are the mean ± standard deviation (SD) of at least four biological replicates conducted at different times.

### 2.7. Plant Material

*Medicago sativa* seeds (cultivar Legend) were purchased from an Italian seed consortium (N. Sgaravatti & C. Sementi S.p.a., Arezzo, Italy) and kept by the Institute of Biosciences and BioResources. To surface sterilize seeds, a modification of the method described by Bucciarelli et al. [37] was used. Seeds were chemically scarificated with concentrated sulfuric acid for 8 min, rinsed with sterile water, and surface sterilized for 15 min with commercial-grade bleach. Seeds were then washed several times with sterilized distilled water and subjected to cold imbibition (48 h at 4 °C in the dark) in petri dishes with two filter papers moistened with 10 mL of sterile water. The petri dishes were kept in the dark at 20 °C for 48 h for germination. Germinated seeds were then transferred into hydroponic units (plastic baskets of a 35 cm length × 15 cm height) containing the nitrogen-free nutrient solution [26]. Infections with rhizobia were performed on germinated seeds by adding 10^4^ cells per root or per mL of hydroponic solution. Each planting unit was kept in the growth chamber under long daylight (16 h), a 19–23 °C temperature, and a 75% relative humidity [26]. Bacteria were isolated from nodules, and their identities were verified by antibiotic resistance patterns. Shoot fresh weight was determined at 21, 28, 35, and 42 days after infection (DAI).

### 2.8. IAA Content

Fresh material (shoots and portion of nodulated roots) was weighted, frozen in liquid nitrogen, and ground to a fine powder. The tissues were immediately homogenized with the extraction solvent [80% (*v*/*v*) methanol, 2% (*v*/*v*) glacial acetic acid, 10 mg/L of butylated hydroxytoluene] and extracted for 24 h, at 4 °C, in the dark. The extracts were dried with the vacuum at 30 °C in the dark and dissolved in phosphate buffer saline (0.01M PBS, pH 7.4). An internal standard (10 nCi) of [*^3^*H]IAA (26 Ci/mmol) (Amersham Radiochemicals, Buckinghamshire, UK) was added before homogenization to estimate the recovery of extracted IAA during the work-up. The displacement immunoassay for IAA determination in leaves and nodulated roots was carried out using the methods described by Bianco and Defez [26]. The results of at least five independent replicates are given in µmol IAA g fresh weight (FW)^−1^.

### 2.9. Quantification of Chlorophylls and Carotenoids

Medicago leaves (0.1 g) at 42 DAI were collected and immediately frozen and ground in liquid N_2_. Chlorophylls and carotenoids were extracted with two steps of ethanol extraction. In the first step, the leaves were incubated in 8 mL of 80% (*v*/*v*) ethanol for 24 h at room temperature and the ethanol solution was then recovered by centrifugation. The resulting particulates were further extracted with 4 mL of 99% (*v*/*v*) ethanol for 24 h. The liquid extract was recovered and combined with that obtained in the first step. The absorbance of the total extract was read at 470, 648.6, and 664.2 nm, and the content of chlorophyll a (C_a_), chlorophyll b (C_b_), total chlorophylls (C_a+b_), and total carotenoids (C_x+c_) was determined as follows: C_a_ = [(13.36 × A_664.2_) – (5.19 × A_648.6_)]; C_b_ = [(27.43 × A_648.6_) – (8.12 × A_664.2_)]; (C_a+b_) = [(5.24 × A_664.2_) + (22.24 × A_648.6_)]; (C_x+c_) = [(1000A_470_ – 2.13C_a_ – 97.64C_b_)/209] [16]. The results are presented as the means ± SD of at least five biological replicates conducted at different times.

### 2.10. Chlorophyll a Fluorescence and Non-Photochemical Quenching Measurements

In vivo chlorophyll a fluorescence, one of the most widely used techniques to examine the photosynthetic performance in plants [21], was measured in Medicago leaves using the Dual-PAM-100 (Walz, Effeltrich, Germany, http://www.walz.com/). Pulses (0.5 s) of red light (5000 mmol·m^−2^·s^−1^) were used to determine the maximum quantum yield of photosystem II (PSII), F_V_/F_M_ (F_m_-F_0_/F_m_); the effective quantum yield of PSII, Y(II); and the non-photochemical quenching (NPQ) [38,39]. Plants were first adapted to the dark for 1 h and the F_V_/F_M_ parameter was measured (photosynthetically active radiation, PAR, equal to zero). The 15-min exposures to different red light intensities (from 6 to 1287 μmol·m^−2^·s^−1^) were used to drive electron transport before measuring the Y(II) parameter. Six plants of *Ms*-0121 and *Ms*-RD64 were analyzed at 21, 28, 35, and 42 DAI, and the average values and standard deviations of F_V_/F_M_, Y(II), and NPQ were calculated.

### 2.11. Acetylene Reduction Assays (ARA)

To analyze the activity of the nitrogenase enzyme, nodulated roots of alfalfa plants at 21, 28, 35, and 42 DAI were cut and transferred into glass tubes (14 mL) that were airtight, with a serum cap. The tubes containing an atmosphere of 10% acetylene were then incubated for 30 min at room temperature [40]. Three samples (1 mL) were taken at different times and the amount of ethylene produced was evaluated using a gas chromatograph (Perkin-Elmer, Clarus 580, Walthan, MA, USA) equipped with a TG-BOND Alumina (Na_2_SO_4_ deactivation) column (30 m × 0.53 mm, Thermo Scientific, Walthan, MA, USA) and a hydrogen flame detector. The carrier gas (helium) flow rate was 48 cm/s, and the oven program was isocratic for 3 min at 130 °C. Activity data are the mean ± SD of at least four biological replicates conducted at different times.

### 2.12. Total C and N Analysis

Total organic carbon and nitrogen contents from 2 mg of Medicago leaves at 42 DAI were determined with dried material by using an elemental analyzer (Fisons EA 1108 CHNS-O Analyzer, Thermo Scientific, Walthan, MA, USA). Data presented are the mean ± SD of at least three biological replicates conducted at different times.

### 2.13. Analysis of Plant-Soluble Proteins

The soluble leaf proteins were extracted in ice cold extraction buffer containing 100 mM Tri-HCl (pH 8.0), 10 mM MgCl_2_, 10 mM NaHCO_3_, 1 mM ethylenediaminetetraacetic acid (EDTA), 12.5% (*v*/*v*) glycerol, 0.1% (*v*/*v*) Triton, 1% (*w*/*v*) insoluble polyvinyl polypyrrolidone, and the complete mini EDTA-free protease inhibitor cocktail (Sigma-Aldrich, St. Louis, MO, USA). The samples were homogenized using a Tissue Lyser (QIAGEN, Hilden, Germany) and the extract was clarified by centrifugation at 13,000× *g* for 30 min at 4 °C. The content of soluble proteins was measured by the Bradford protein assay, with the BSA as a standard. The supernatants were stored at −20 °C prior to SDS-PAGE and immunoblot analysis. The proteins were separated by 12.5% SDS-PAGE with a Mini Protean II cell (BIO-RAD, Hercules, CA, USA), according to Laemmli [41]. After electrophoresis, proteins were blotted onto PVDF membranes, according to the standard procedures. Blots were probed with anti-RbcL (Rubisco large subunit, AGRISERA, Vännäs, Sweden) and anti-rabbit IgG AP-conjugated (BIO-RAD, Hercules, CA, USA) as secondary antibodies and developed with nitroblue tetrazolium and 5-bromo-4-chloro-3-indolyl phosphate. Developed immunoblots were scanned using the VersaDoc imaging system (BIO-RAD, Hercules, CA, USA) and processed using the Quantity One software (BIO-RAD, Hercules, CA, USA). For immunoblot quantification, RbcL protein standard (AGRISERA, Vännäs, Sweden) was subjected to SDS-PAGE and immunoblotting. Electrophoresis and immunoblotting were repeated with four different biological replicates and one representative picture of the results is given.

### 2.14. GC-MS Analysis

Leaf samples (50 mg) were collected from Medicago plants at 42 DAI and immediately stored at −20 °C until further analysis. Leaf samples were homogenized in 1 mL extraction buffer A (methanol/chloroform/water 2.5:1:1) containing 1 μmol Norleucine (NOR, Sigma Aldrich, St. Louis, MO, USA) as an internal standard by using a TissueLyser (QIAGEN, Hilden, Germany) at 30 cycles per second for 5 min at 4 °C. Samples were centrifuged at 16,000× *g* and 4 °C for 10 min. A second extraction step was performed using 0.5 mL extraction buffer B (methanol/chloroform 1:1) in a rotating mixer at 4 °C for 30 min. Samples were centrifuged at 13,500× *g* and 4 °C for 5 min. Supernatants of the first and second extractions were combined and 0.25 mL water (MilliQ) was added to aid the phase separation. Samples were mixed rigorously for 1 min and then centrifuged at 13,500× *g* for 5 min. The upper phases were collected and dried in a centrifugal vacuum concentrator (Savant Speedvac, Eppendorf, Hamburg, Germany). Dried leaf extracts were dissolved in 50 μL of methoxyamine hydrochloride (MOX, Sigma Aldrich, St. Louis, MO, USA) in pyridine (20 mg/mL, *w*/*v*), rigorously mixed (vortexed), and sonicated for 1 min in an ultrasonic bath. MOX reacts with both aldehydes and ketones to form oxim methyl ethers and thus prevents multiple derivatives when enols are present during silylation. The derivatization reaction was left to proceed at 30 °C for 90 min. The reaction was followed by trimethylsilylation, which included adding 50 μL of N,O-Bis(trimethylsilyl)trifluoroacetamide (BSTFA) and trimethylchlorosilane (TMCS) 99:1 (Supelco, Sigma-Aldrich, St. Louis, MO, USA) to the samples, vortexing them, and keeping them at 37 °C for 60 min. Samples were centrifuged at 10,000× *g* at 22 °C for 5 min. Supernatant was diluted 10 times using freshly prepared MOX:BSTFA = 1:1. Samples (1 μL) were injected into the gas chromatography (GC) column using the 1:10 split mode. Amino acid standards, l-alanine, l-serine, l-valine, dl-threonine, and l proline (Calbiochem, San Diego, CA, USA), were dissolved in water, and l-glutamic acid and DL-aspartic acid (Calbiochem, San Diego, CA, USA) were dissolved in 0.1 N HCl at 10 mg/mL and used to prepare a mix of equimolar amino acid solution at 1000 μM. Single calibration solutions were prepared by diluting the stock solution and adding 1 μmol internal standard NOR. Standards were dried out and derivatized according to the plant material. For gas-chromatography-mass spectrometry (GC-MS) analysis, the Trace 1300 GC coupled to the TSQ DUO triple quadrupole mass spectrometer (Thermo Scientific, Walthan, MA, USA) was used. The derivatized samples were separated using a DB-5 column (30 m length, 0.25 mm internal diameter, 0.25 μm film, Thermo Scientific, Walthan, MA, USA). The GC oven conditions were as follows: initial temperature 70 °C hold 1 min, ramp-1 1 °C/min to 76 °C, ramp-2 6 °C/min to 200 °C, ramp-3 40 °C/min to 325 °C, and hold 5 min. Helium was used as a carrier gas at a 1.2 mL/min constant flow. The total run time was 40 min. The transfer line and ion source temperatures were 240 and 250 °C, respectively. Electron impact ionization was used at 70 eV. Samples were acquired using two different scan modes: full scan and selected reaction monitoring (SRM). The acquisitions with both scan modes were carried out in triplicate. Full scan acquisitions were performed in the range of *m/z* 40–600, with a scan time 250 ms. Compounds were identified through mass spectral matching using the NIST database and applying match cut-off criteria of 700/1000. Conditions for the collision induced dissociation (CID) ion transmissions of the single amino acids (Appendix A) were determined using the Thermo Scientific AutoSRM application run under Chromeleon 7.2 software package. Argon was used as a collision gas. Concentrations of amino acids in the samples were calculated using external calibration curves for each amino acid and values were corrected against the internal standard. For organic acids, relative response ratios were calculated by normalizing the respective peak areas to the peak area of the internal standard. For soluble sugars, the relative abundances were measured in *Ms*-1021 and *Ms*-RD64 plants. To obtain relative abundances, the peak areas of different trimethylsilyl (TMS derivatives of the sucrose, fructose, and glucose were summed and divided by the peak area of the internal standard. Mean normalized values were obtained for six biological replicates.

## 3. Results

### 3.1. Endogenous Biosynthesis of IAA does not Induce Genomic Changes

To rule out the possibility that the changes introduced by the IAA overexpression in RD64 cells led to genetic variations that may have been selected during the initial phases of the symbiotic process, genomic DNA of both the 1021 and the RD64 bacteroids was subjected to whole genome sequencing. The workflow described in Appendix A was used to localize genomic differences between the wild-type *E. meliloti* 1021 and the modified RD64 strains under a symbiotic condition. Genomic DNA of both the 1021 and the RD64 bacteroids, isolated from seven-week-old nodules, was subjected to whole genome sequencing, by using the Illumina HiSeq2000 platform. Each resulting sequence was mapped onto the genome of the free-living *Ensifer meliloti* 1021 (ASM696v1) downloaded from NCBI (ftp://ftp.ncbi.nlm.nih.gov/genomes/Bacteria/Sinorhizobium_meliloti_1021_uid57603/).

After the filtering process (Materials and Methods section), only mutations obtained by at least two algorithms were considered for validation through direct PCR and Sanger sequencing. Comprehensive lists of the specific mutations (Appendix A) and primers flanking each SNV (Appendix A), used for PCR, are included as Appendix A. PCR and Sanger sequencing analyses were applied on three different genomic 1021 and RD64 DNA samples to validate sixteen variants. For almost all the selected mutations, no dominant signal peaks, indicating the mutation for both 1021 and RD64 samples, were observed. Four of them (SMc01138, SMb21229, SMa0233, and SMa2061) were confirmed in only one RD64 sample, but dominant peaks in the sequencing trace were also observed at a lower level in one 1021 sample (Appendix A). Such results led us to consider these variants as false positives. Excluding common variation, no specific genetic change was observed for RD64 bacteroids compared to 1021 ones, meaning that the endogenous biosynthesis of IAA did not induce genomic changes.

### 3.2. IAA Influences the Expression of Bacterial Genes within Root Nodules

To check whether the IAA-overproduction inside nodules of Medicago plants infected with RD64 had an effect on the gene expression in bacteroids, the transcript levels of selected genes were measured in root nodules of RD64-nodulated plants and compared with the ones estimated for plants nodulated by the wild-type strain 1021. For each gene, the 2^−∆∆Ct^ values refer to the relative fold differences in RD64-nodulated plants compared to 1021-nodulated ones. Almost all the analyzed genes were selected from the list of genes up- or down-regulated in *E. meliloti* RD64 free-living cells, and were involved in nitrogen-fixation, metabolism, the stress response, and DNA repair [35]. Among the nitrogen-fixation genes, there was *fixJ*, coding for the response regulator that turns on nitrogen-fixation genes, as well as other ones transcriptionally activated by the *fixJ* gene product. The expression level of *fixJ* increased over time, reaching its highest level at 42 DAI (Figure 1A). A similar trend was observed for the genes *nifA*, *fixK1*, and *fixK2*, directly regulated by the FixJ protein, and for *nifH*, coding for the nitrogenase enzyme. Accordingly, the gene *fixA*, which is part of the *fixABCX* transcriptional unit involved in the electron transport to nitrogenase [42], began to be significantly induced at 28 DAI, reaching its highest expression at 42 DAI. At the last time point, the highest increase was measured for *fdxB*, coding for the ferredoxin, which acts as an electron donor for dinitrogenase reductase [43]. The expression level of the *fixNOQP*_1,2_ operon genes, coding for heme-copper *cbb3*-type oxidases with a high affinity for oxygen [29] (Figure 1A), was also evaluated. A moderate increase in the expression of the copy 2 of the gene *fixN* was measured at all analyzed time points, with a slightly higher induction recorded at 42 DAI. For the copy 1 of the *fixNOQP* operon, the increased expression of all four genes began to be significant at 35 days and reached the highest induction at 42 DAI. These results were in agreement with those reported in other works [44] and suggest, contrary to the observation made by Torres et al. [45], that the copy 1 of the *fixNOQP* operon is important for nitrogen fixation, mostly during the late stages of the symbiosis. We also found that the transcript levels of *rpoH1*, coding for one of the two alternative s^32^ factors present in the *S. meliloti* genome, significantly increased. This sigma factor operates under heat shock and oxidative stress and is required for effective nitrogen-fixing symbiosis with alfalfa [46]. When the transcript levels of the auxin response gene *GH3* [47] were evaluated, we verified that the expression of this gene was strongly increased in the root of RD64-nodulated plants. Considering that all genes described above showed the highest induction levels at 42 DAI, the expression of genes involved in metabolism and genetic information processing has been evaluated at 42 DAI (Figure 1B). The genes coding for the enzymes involved in the tricarboxylic acid (TCA) cycle and glyoxylate shunt cycle were among the selected metabolic genes. We verified that the expression of *gltA*, *icd*, and *sucA*, coding for the key TCA cycle enzymes, was significantly increased in RD64 bacteroids (Figure 1B). The transcript levels of *aceA* and *glcB*, involved in the glyoxylate shunt cycle, did not significantly change (Figure 1B). Finally, the expression of two components of the nucleotide excision repair (NER) system, which recognize and remove a large number of DNA lesions induced by a wide variety of environmental agents [48], was evaluated. We found that the expression of the genes *uvrA* and *uvrB*, the two components that first interact with DNA, significantly increased in RD64 bacteroids (Figure 1B).

### 3.3. The IAA-Overproducing E. Meliloti RD64 Strain Improves the Phenotype of the Medicago Host Plant

Different physiological parameters were evaluated for Medicago plants nodulated by the IAA-overproducing RD64 strain (*Ms*-RD64) and the data obtained were compared with those recorded for plants nodulated by the wild-type strain 1021 (*Ms*-0121).

#### 3.3.1. IAA Analysis

It has previously been demonstrated by TMS GC-MS analysis that the expression of the chimeric operon p-*iaaMtms2* in *E. meliloti* bacteroids (strain RD64) resulted in a 10-fold increase of IAA content in the root nodules of *M. sativa* (the IAA concentration measured for nodules of *Ms*-RD64 was 0.12 nmol/g FW, whereas the one estimated for nodules of *Ms*-1021 was 1.2 nmol/g FW) [49]. The ELISA immunoassay previously developed to measure the IAA levels in *M. truncatula* plants nodulated by the *E. meliloti* strains RD64 and 1021, was used here to estimate the concentration of IAA in different *M. sativa* tissues. This analysis revealed that the leaves of 28-day-old *Ms*-1021 and *Ms*-RD64 plants had almost the same IAA content (Table 1). In contrast, when portions of root-containing nodules were analyzed, the IAA level measured for *Ms*-RD64 plants was more than three-fold higher than the one measured for the *Ms*-1021 plants. This result was consistent with the ability of RD64 to produce up to 93-fold (56 µM) more IAA than its parental strain 1021 (0.63 µM) under free-living conditions [25].

#### 3.3.2. Nitrogenase Activity

The symbiotic phenotype of alfalfa plants nodulated by the wild-type strain 1021 and its IAA-overproducing derivative RD64 was studied under hydroponic growth conditions. Medicago plants inoculated with the modified strain RD64 showed a significant increase in acetylene reduction activity compared to those nodulated by the wild-type strain 1021 (Figure 2A). This effect was particularly evident at 35 and 42 days after infection (DAI).

#### 3.3.3. Total N and C Content

The dry combustion with an elemental analyzer allowed the measurement of a significant increase (more than 35%) in both the N and C content for *Ms*-RD64 plant leaves (0.025 ± 0.001 g N/plant; 0.285 ± 0.008 g C/plant) compared to *Ms*-1021 plants (0.018 ± 0.001 g N/plant; 0.208 ± 0.001 g C/plant) at 42 DAI. However, no changes were observed for the C/N ratio.

#### 3.3.4. Pigment Determination

The chlorophyll content in leaves is proportional to the intensity of nitrogen fixation and depends on the symbiotic properties of bacteroids inside root nodules; this means that the photosynthesis provides energy for nitrogen reduction [6,10,50]. Here, we report that the leaves of RD64-inoculated plants contained a higher amount of chlorophyll pigments compared to those nodulated by the wild-type strain (Figure 2B). In particular, at 42 DAI, an increase of 37% and 84% for chlorophyll a (the major pigment) and chlorophyll b (the accessory pigment) was measured, respectively. The ratios of chlorophyll a to chlorophyll b registered for *Ms*-RD64 and *Ms*-1021 plants were 3.7 ± 0.6 and 2.7 ± 0.2, respectively, meaning that the light adaptation/acclimation of their photosynthetic apparatuses was different. The carotenoid amount, the well-known light-harvesting pigments involved in the protection of photosynthetic apparatus [20], was also measured. *Ms*-RD64 plants produced up to 72% more total carotenoids (4.0 ± 0.3 mg/g DW) than *Ms*-1021 plants (2.9 ± 0.5 mg/g DW). The positive correlation observed in *Ms*-RD64 plants for the chlorophyll and carotenoid content might indicate that these plants were more protected against light-induced damage.

#### 3.3.5. Photosystem II (PSII) Activity in Response to Light

The effective PSII quantum yield [Y(II)] and the light-induced NPQ generation during illumination with a red light intensity up to 1287 μmol·m^−2^·s^−1^ were measured at 42 DAI. In the range from 20 to 500 μmol·m^−2^·s^−1^ light intensities, a strong decline in Y(II) was recorded per unit leaf area, whereas at the highest light intensities (530–1200 μmol·m^−2^·s^−1^), a weaker decrease was registered for both *Ms*-1021 and *Ms*-RD64 plants (Figure 2C), thus indicating that the two samples had a comparable photosynthetic performance. When the light-induced NPQ generation was measured in Medicago leaves, a significant increase was observed for *Ms*-RD64 plants (Figure 2D) compared those of *Ms*-1021. The maximum difference was reached in the range of 200 and 400 μmol·m^−2^·s^−1^. Such an increase could be associated with an improved efficiency in removing excess excitation energy within chlorophyll-containing complexes and was consistent with the greater amount of carotenoids produced by these plants. 

### 3.4. Soluble Proteins and Rubisco Content

When the soluble protein levels extracted from *Ms*-RD64 and *Ms*-1021 leaves were measured and compared over time, the following trend was observed: at 21 DAI, the total protein level per plant was lower in *Ms*-RD64; at 28 DAI, no significant difference was observed; and at 35 and 42 DAI, a significantly greater amount was measured for *Ms*-RD64 plants (Figure 3A). The content of the most abundant plant-soluble protein, Rubisco, the enzyme widely considered as the rate-limiting step in photosynthetic carbon fixation [12], was analyzed using polyclonal antibodies against Rubisco’s large subunit (RbcL). A significant increase in RbcL quantity was observed over time for *Ms*-RD64 plants compared those of *Ms*-1021. At 42 DAI, the RbcL band intensity was about 37% higher in *Ms*-RD64 plants (Figure 3B,C). This increase was consistent with the one measured for the total soluble proteins and confirmed the previously published data [51]. Rubisco is relatively inefficient and a large amount of this enzyme is needed to support carbon fixation. When the RD64- and 1021-nodulated plants were compared, the higher Rubisco content measured for RD64-nodulated plants correlated with the increased C content measured for these plants.

### 3.5. Metabolomic Profiles

Metabolic analysis was carried out on the leaves of *Ms*-1021 and *Ms*-RD64 plants to measure the free-amino acid content and to monitor the presence of some other metabolites, including organic acids and soluble sugars. Quantitative analysis of the free amino acids showed that valine, alanine, proline, aspartic acid, and glutamic acid significantly increased in *Ms*-RD64 plants compared those of *Ms*-1021 (Figure 4A,B). In particular, aspartic acid and proline increased by 172% and 80%, respectively. In contrast, no significant differences were observed for serine and threonine. Similarly, a higher proline concentration was found when *Ms*-RD64 plants were subjected to controlled osmotic drought stress [51], and when a different Medicago genotype was analyzed under salt-stress conditions [26]. The increase in the nitrogen-rich amino acids level was connected to the accumulation of Rubisco protein observed for *Ms*-RD64 plants and was well-related to the enhanced nitrogenase activity measured for these plants, indicating that the N availability is not a limiting factor. Other metabolites were identified through mass spectral matching using the NIST database (Table 2). Considerable differences were observed between the GC-MS metabolomic profiles of *Ms*-1021 and *Ms*-RD64 plants. To obtain a crude estimate of the relative abundance of some metabolites, relative chromatographic peak areas against the internal standard peak area were calculated (analyte peak area/IS peak area) and considered as the “response”. *Ms*-RD64 plants showed a higher level of sucrose compared to *Ms*-1021 (Table 2). The two primary components of sucrose, fructose and glucose, were also accumulated in these plants (Table 2). The analysis of organic acids showed that the abundance of two C_4_-dicarboxylates, malate and citrate, derived through sucrose metabolism, increased in *Ms*-RD64 plants (Table 2). The increase of the malate level, one of the main substrates needed by bacteroids for ATP generation, was consistent with the induction of nitrogenase activity observed for these plants. A similar increase in malate production was measured when RD64 free-living cells, presenting more active TCA cycle rate-controlling enzymes, were grown on minimal medium containing an insoluble phosphate source [27].

### 3.6. Plant Yield

When the shoot fresh weights of *Ms*-1021 and *Ms*-RD64 plants were compared, a biomass increase over time was observed for *Ms*-RD64 plants, with the highest increment (62% higher) being reached at 42 DAI (Figure 4C). These data were connected to the different nodules’ morphology previously observed for Medicago plants infected with the IAA-overproducing rhizobia [24,25]: the meristematic zone I and the interzone II-III [52] were up to 70% and 30% more extended, respectively, than plants nodulated by the wild-type strain.

## 4. Discussion

In this study, the plant growth-promoting activities of the IAA-overproducing strain *E. meliloti* RD64 and its parental wild-type strain 1021 were compared to evaluate whether the RD64-nodulated plants showed alteration in the main metabolic processes. *M. sativa* was used as the model host plant and genetic, metabolic, and physiological analyses were carried out during its initial life cycle. The RD64 strain was introduced and described for the first time in Imperlini et al. [24]. Here, we show that the bacteroids deriving from RD64 and its parental strain 1021 are isogenic, as the IAA-overproduction in RD64 cells does not cause mutations in the few bacteria that succeed in overcoming the bottleneck of symbiosis via the root hairs and infection treads. We provide evidence that the transcript levels of genes involved in nitrogen fixation, including *nifH*, energy metabolism, and the stress response, increased inside nodules of RD64-nodulated plants compared to the plants nodulated by the wild-type and poor IAA-producer strain 1021. The highest positive effect was observed at 42 DAI. These data were connected to the increase in the nitrogenase activity and to the improved stress response measured for these plants. Positive effects on the expression of these genes were also observed when the transcriptional profiles of IAA-overproducing *E. meliloti* RD64 cells were analyzed [35]. The data presented also show that IAA was able to induce, within infected nodule tissue, the expression of respiratory oxidase genes, as already observed under free-living conditions [35]. The higher transcript levels measured for these genes allowed the formation of a suitable microaerobic environment, required for the expression of nitrogen-fixation genes. Here, we report that the *Ms*-RD64 plants accumulated greater amounts of pigments (chlorophylls and carotenoids), nitrogen-rich amino acids (such as proline, aspartic acid, glutamic acid, and alanine), and soluble proteins, including Rubisco, catalyzing the first major step of carbon fixation. The chlorophyll *a* fluorescence analysis revealed that the fluorescence emission per unit leaf area measured for *Ms*-RD64 and *Ms*-1021 plants did not change. These findings support the evidence that, in most species, substantial increases in crop yield do not correspond to significant changes in the rate of photosynthesis per unit leaf area [53,54]. In contrast, the fraction of energy dissipated as heat was higher in *Ms*-RD64 plants. The combination of higher non-photochemical quenching and a greater amount of carotenoids measured for these plants could prevent the likelihood of the formation of damaging free radicals. The over-investment in the Rubisco protein and nitrogen-rich amino acids observed for *Ms*-RD64 plants might have contributed to the more efficient stress response observed for these plants [26,27,51]. Indeed, Rubisco provides a means of storing N that can be utilized by the plants’ reproductive components when Rubisco degradation is initiated during stress conditions and senescence [55,56,57]. The increase in total carbon content and organic acids observed for *Ms*-RD64 plants was consistent with the higher level of storage compounds, such as poly-β-hydroxybutyrate, accumulated in the more extended interzone II-III of their root nodules [24]. In addition, the greater amount of end-products of photosynthesis, such as glucose, fructose, and sucrose, measured for these plants indicates that they had a more effective energy source to be used for their growth. The significant biomass increase measured for *Ms*-RD64 plants could be linked to the accumulation of such metabolic products. In legume plants, a substantial part of the energy input accumulated in the photosynthetic process is used to feed the energy-consuming nitrogen fixation. The overall data reported in this study show that the more efficient nitrogen-fixing apparatus of RD64-nodulted plants stimulates C assimilation through photosynthesis. The enhancement of these two processes occurred in a coordinated manner: the absolute levels of C and N significantly increased, but their relative ratios did not change when the *Ms*-1021 plants were used as a reference. The enhancement of plant growth promotion observed for RD64-nodulated plants could be related to this effect.

## 5. Conclusions

In recent years, several studies have shown that the use of PGPR represents an environmentally sound way of increasing the crop yield by stimulating plant growth through direct or indirect mechanisms. However, due to the properties often inconsistent with the inoculated PGPR, the use of PGPR in the agricultural industry is still limited. Considering that, in a sustainable agricultural system, crops need to overcome disease resistance, different abiotic stresses, and nutritional deficiency, an ideal PGPR should possess a broad spectrum of action. Here, we show that by using a simple system to increase the endogenous IAA biosynthesis in rhizobia strains, a PGPR with different capacities can be obtained. The results observed for Medicago plants nodulated by the IAA-overproducing strain RD64, lead us to speculate that PGPR stimulating the C and N metabolisms in a balanced manner could be used to significantly increase the plant yield.

## Figures and Tables

**Figure 1 microorganisms-07-00403-f001:**
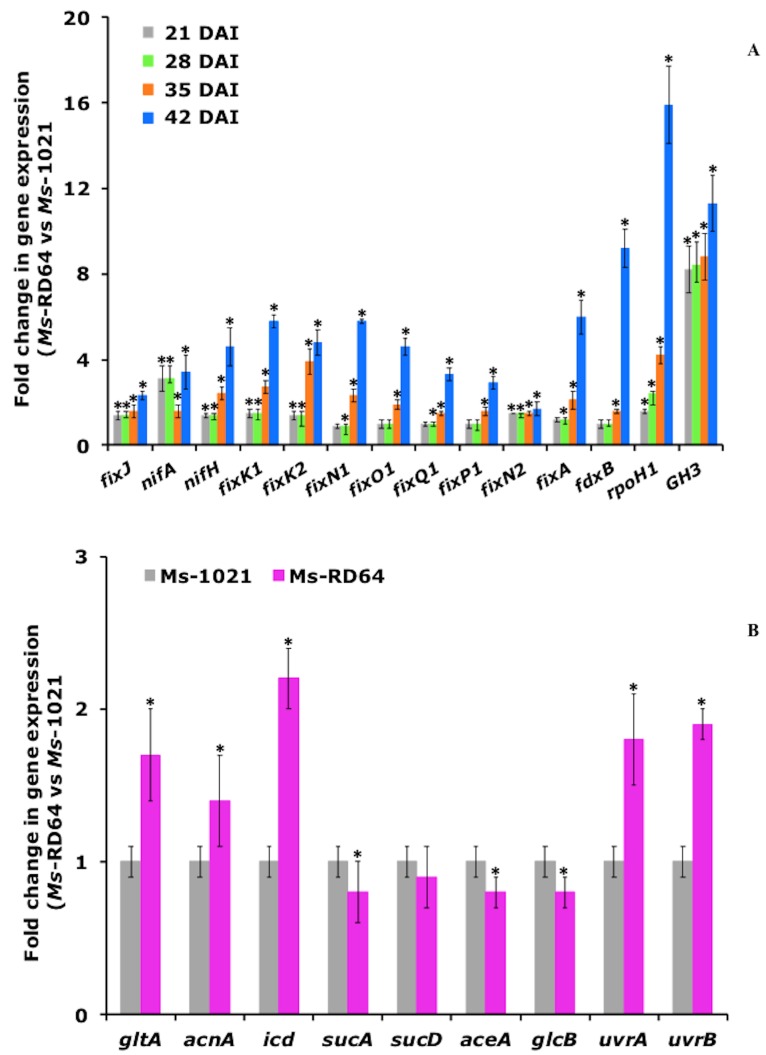
The expression of genes involved in nitrogen-fixation, energy metabolism, stress response, and genetic information processing increased in Medicago plants nodulated by *Ensifer meliloti* RD64. (**A**) Quantitative measurement of transcript levels for nitrogen-fixation genes, *rpoH1* and *GH3*, in root nodules of Medicago plants nodulated by the strains *E. meliloti* RD64 and 1021 at 28, 21, 35, or 42 days after infection (DAI). (**B**) Quantitative measurement of transcript levels for energy metabolism and DNA repair genes in root nodules of Medicago plants nodulated by the strains *E. meliloti* RD64 and 1021 at 42 DAI. Fold change >1: genes more highly expressed in Medicago plants nodulated by RD64; fold change <1: genes more highly expressed in Medicago plants nodulated by 1021. Values are the means ± standard deviation (SD) of four different biological replicates. The asterisks indicate significant differences (*P* < 0.05, one-way ANOVA with Tukey’s post-hoc test) between 1021- and RD64-nodulated plants.

**Figure 2 microorganisms-07-00403-f002:**
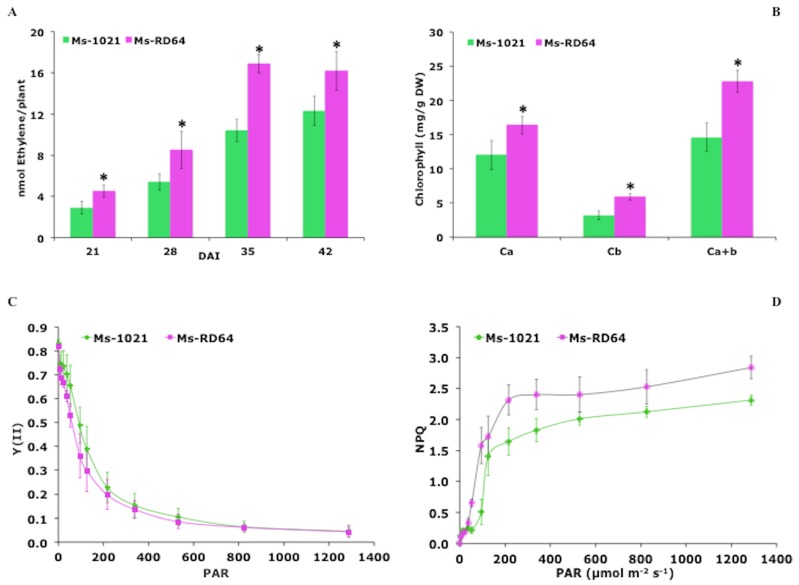
Medicago plants nodulated by the IAA-overproducing *Ensifer meliloti* RD64 show significant changes in nitrogenase activity, chlorophyll content, and light-induced non-photochemical quenching (NPQ) generation. (**A**) The activity of the nitrogenase enzyme was measured by the acetylene reduction assay (ARA) on root sections of nodulated Medicago plants harvested at different times after inoculation. Data presented are the mean ± SD of at least five replicates. (**B**) Chlorophyll content extracted from leaves at 42 days after infection (DAI). (**C**) Light-dependent changes of the effective PSII quantum yield [Y(II)] for dark-adapted plant leaves at 42 DAI. (**D**) NPQ for dark-adapted plants leaves at 42 DAI. Data presented are the mean ± SD of six biological replicates. The asterisks in panel A and B indicate significant differences (*P* < 0.05, one-way ANOVA with Tukey’s post-hoc test) between 1021- and RD64-nodulated plants.

**Figure 3 microorganisms-07-00403-f003:**
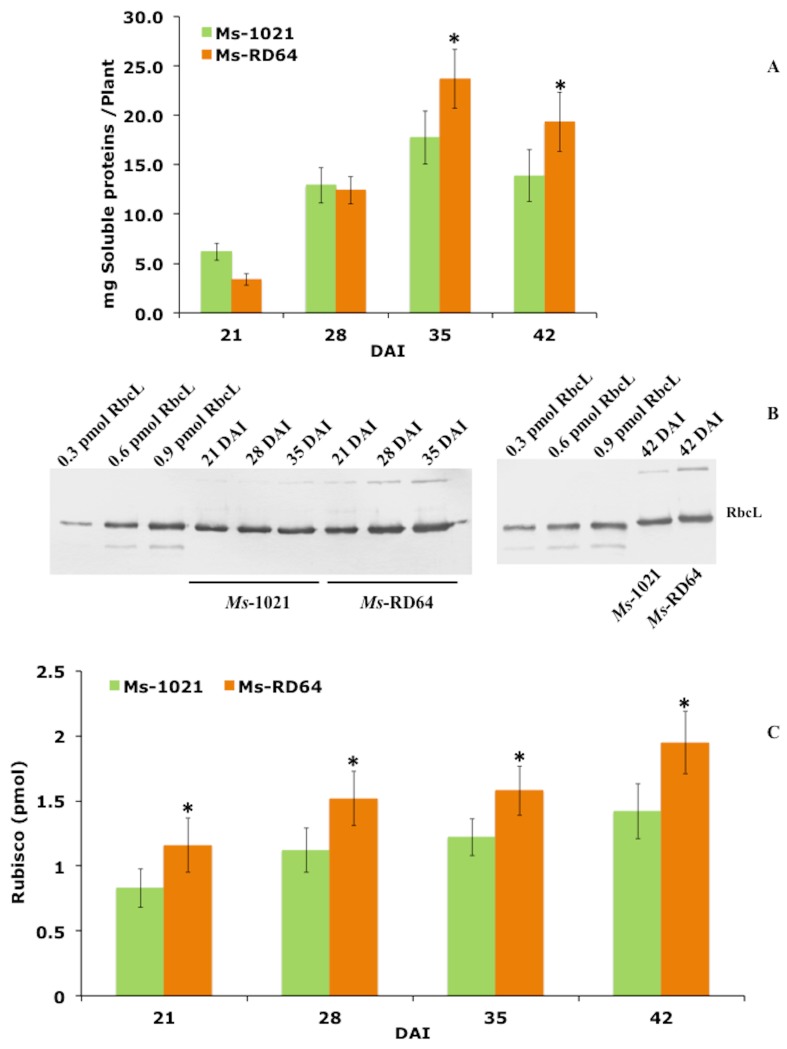
IAA stimulates the biosynthesis of the most abundant plant-soluble protein, Rubisco, in nodulated Medicago plants. (**A**) Soluble protein content in nodulated Medicago plants over time. (**B**) Immunoblot analysis of extracts from *M. sativa* leaves collected at 28, 21, 35, or 42 days after infection (DAI) subjected to SDS-PAGE, transferred to a nitro-cellulose membrane, and probed with anti-RbcL antibodies (RbcL, Rubisco’s large subunit). (**C**) Immunoblot quantification of the RbcL protein. Data presented are the mean ± SD of four biological replicates. The asterisks in panels A and C indicate significant differences (*P* < 0.05, one-way ANOVA with Tukey’s post-hoc test) between 1021- and RD64-nodulated plants.

**Figure 4 microorganisms-07-00403-f004:**
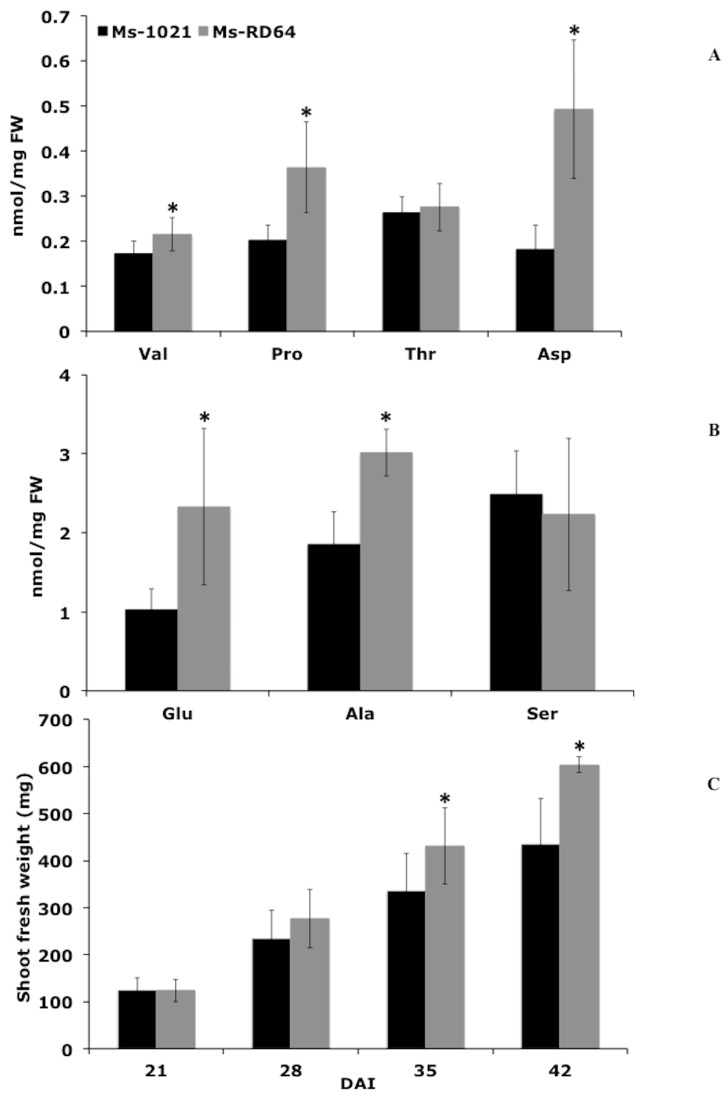
The inoculation of Medicago plants with the IAA-overproducing *Ensifer meliloti* RD64 increases the biosynthesis of nitrogen-rich amino acids and promotes plant growth. (**A**) Valine (Val), proline (Pro), threonine (Thr), and aspartic (Asp) content. (**B**) Glutamic (Glu), alanine (Ala), and serine (Ser) content. Data presented are the mean ± SD of seven biological replicates. (**C**) Shoot fresh weight of nodulated Medicago plants measured at 21, 28, 35, and 42 days after infection (DAI). Data presented are the mean ± SD of thirty biological replicates. The asterisks in all the panels indicate significant differences (*P* < 0.05, one-way ANOVA with Tukey’s post-hoc test) between 1021- and RD64-nodulated plants.

**Table 1 microorganisms-07-00403-t001:** Indole-3-acetic acid (IAA) levels in *Medicago sativa* plants nodulated by the strains 1021 and RD64.

Sample	IAA Content (µmol *g* FW^−1^)	Ratio	*P* Value
*Ms*-1021 *Leaf*	60.1 ± 6.5		
*Ms*-1021 *Root*	19.4 ± 2.2	0.3	<0.01
*Ms*-RD64 *Leaf*	59.7 ± 7.4		
*Ms*-RD64 *Root*	63.2 ± 11.1	1.1	<0.01

Leaves and roots of 28-day-old plants were collected and immediately used for IAA extraction and quantification. Data presented are the mean ± standard deviation (SD) (*n* = 7). Ratio calculation was based on the IAA content of the leaf tissue. The significance of the ratio (*P* < 0.01) was confirmed by Student’s *t* test.

**Table 2 microorganisms-07-00403-t002:** Differences in the production of some metabolites in Medicago plants nodulated by 1021 and RD64 strains.

Compound	^§^Relative Abundance (*Ms*-RD64/*Ms*-1021)
Malic acid	1.8
Citric acid	2.9
Sucrose	1.3
Fructose	1.9
Glucose	1.3

^§^ Relative metabolite abundances measured in *Ms*-1021 and *Ms*-RD64 plants. To obtain relative abundances, the peak areas of different TMS derivatives of the sucrose, fructose, and glucose were summed and divided by the peak area of the internal standard. Mean normalized values were obtained for four biological replicates. The significance of the ratios (*P* < 0.05) was confirmed by Student’s *t* test.

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
