# Peer review of "Bacterial IAA-Delivery into Medicago Root Nodules Triggers a Balanced Stimulation of C and N Metabolism Leading to a Biomass Increase"

_microorganisms, 2019, doi:10.3390/microorganisms7100403_

Round 1

Reviewer 1 Report

Microorganisms

REVIEW

“Bacterial IAA-delivery into Medicago root nodules triggers a balanced stimulation of C and N metabolism leading to biomass increase” by R. Defez et al. The authors described that RD64 nodulated Medicago sativa plants showed the increasement of plant growth with a balanced C and N ratio in metabolism. In legume plants, symbiosis with nitrogen-fixing rhizobia have been broadly studied. Recently, mamy papers have published that the plant growth promoting rhizobacteria related to IAA, as a biofertilizer, affected crop growth. This ms represented very well that medicago root nodules hosting RD64, a derivative of the E. meliloti engeneered to overproduce the auxin IAA showed increased plant biomass. Therefore, the authors need to address the following minor comments.

The authors need to address the following comments.

Q1)  pp.7 line 31. You should be show morphology differencies between Ms-RD 64 plants and wild-type plants by pictures or tables.

MINOR POINTS 1) In Materials and Methods, they ar e too enough. It would be better to reduce or eliminate the usual methods. For examples, pp. 4 line 4.   2.6 Quantititative real-time PCR analysis,
            pp. 5 line 22.  2.13 SDS-PAGE, immunoblot analysis and protein quantification
            pp. 5 line 10.  2.14 Determination of free amino acids, soluble sugars and organic acids from  
                             Medicago leaves by GC-MS

2. “Table 3. Primers used in qRT-PCR analysis should be moved to Supple data.

Author Response

“Bacterial IAA-delivery into Medicago root nodules triggers a balanced stimulation of C and N metabolism leading to biomass increase” by R. Defez et al. The authors described that RD64 nodulated Medicago sativa plants showed the increasement of plant growth with a balanced C and N ratio in metabolism. In legume plants, symbiosis with nitrogen-fixing rhizobia have been broadly studied. Recently, mamy papers have published that the plant growth promoting rhizobacteria related to IAA, as a biofertilizer, affected crop growth. This ms represented very well that medicago root nodules hosting RD64, a derivative of the E. meliloti engeneered to overproduce the auxin IAA showed increased plant biomass. Therefore, the authors need to address the following minor comments.

The authors need to address the following comments.

Q1)  pp.7 line 31. You should be show morphology differencies between Ms-RD64 plants and wild-type plants by pictures or tables.

Reply – The evidence you required have been already reported in Imperlini et al. [24] and Bianco et al. [25]. Both references have been cited in the Introduction section.

MINOR POINTS 1) In Materials and Methods, they are too enough. It would be better to reduce or eliminate the usual methods. For examples, pp. 4 line 4.   2.6 Quantititative real-time PCR analysis,
pp. 5 line 22.  2.13 SDS-PAGE, immunoblot analysis and protein quantification
pp. 5 line 10.  2.14 Determination of free amino acids, soluble sugars and organic acids from 
                             Medicago leaves by GC-MS

Reply – We followed your suggestions and modified the text accordingly.

“Table 3. Primers used in qRT-PCR analysis should be moved to Supple data.

Reply – Further to your suggestion the Table 3 has been moved to the Supplementary Materials.

Reviewer 2 Report

The basic premise of this manuscript is that rhizobacteria that overproduce indole-3-acetic acid have the capacity to increase metabolic activity resulting in biomass increase.   As the authors note, this has been looked at previously by a number of different approaches and with different results, leading to ambiguous interpretations.  Resolution of these issues would be important.

This present manuscript is difficult for reviewers because of the incessant use of… “according to manufacturer’s instructions” and “determined as described by…” in the place of actually telling the reader what they did.  This can be anything from just annoying to highly irritating (like the Lichtenthaler reference where a huge chapter is cited, but I assume they only needed to point to Table III – or better they could have said succinctly what they did then give the citation! Also, the Bucciarelli et al paper that you have to look up to learn “chemically scarified with concentrated sulfuric acid for 8 min and surface sterilized for 3 min with commercial-grade bleach”).  Of course, an important reviewer task is to ask if the methods were suitable, so tracking down a dozen or so citations is a pain especially given that our time is a free gift to the authors and the journal.

Moving from that aspect to a more serious issue is the methods section on “Determination of free amino acids, soluble sugars and organic acids” where the section is almost entirely devoted to amino acid analysis.  The procedure used is not the best (Artifacts in Trimethylsilyl Derivatization Reactions and Ways to Avoid Them, https://littlemsandsailing.files.wordpress.com/2014/04/silyl_2014_text.pdf) , but is acceptable for amino acids.  Soluble sugar analysis is barely described in this present manuscript and typically TMS reagents react poorly to sugars, often giving multiple peaks and missing sugars in complex plant samples (Zarate et al. 2017 doi:10.3390/metabo7010001). 

The authors state “To analyse the effect of IAA-overproduction inside RD64 bacteroids on the expression of genes involved in nitrogen fixation, qRT-PCR analysis of selected genes was carried out on root nodules of Medicago plants infected with the strains E. meliloti RD64 and 1021 at different DAI.”  There is a curious omission here in that they examined ‘selected genes’ important for nitrogen metabolism, but omitted genes that would have confirmed or negated their central hypothesis regarding IAA “overproduction” – namely they did not look for elevation in the   IAA, GH3, and SAUR families, as well as several PIN genes (Paponov et al. 2008 https://doi.org/10.1093/mp/ssm021).  Such analysis might have helped to understand if indeed additional IAA production is being perceived by the plant auxin signaling systems.  Without this information, the association of the general metabolic responses noted with elevated auxin is mostly just correlation and a bit of conjecture.

They have a problem with history.  They state “The main plant auxin, IAA, was identified in 1937” but from plants this is usually attributed to Berger and Avery in 1944, although it was identified from urine and called ‘heteroauxin’ a decade earlier.  Likewise, many of the earlier studies of indole metabolism altered ‘brown rhizobia’ and of IAA involvement go uncited (see, for example, Williams and Signer 1990).  They also note “…plant hormone biotechnology has been to modulate their internal concentration by modification of either synthesis or degradation pathways, or both [2].” They seem to ignore the classic work of Glass and Kosuge on the  indole-3-acetic acid-lysine synthetase (iaaL gene) (Spaepen and Vanderleyden, 2010  doi: 10.1101/cshperspect.a001438) that is a very effective method for alteration of IAA levels in planta (Romano et al. 1991 DOI: 10.1101/gad.5.3.438). 

Finally, and importantly, a manuscript on the effect of indole-3-acetic acid on plant yield needs to show with some chemical certainty that IAA levels are indeed elevated.  Very few labs use immunoassays for IAA levels in plants, as noted in their review (Tivendale and Cohen 2015 DOI: 10.1007/s00344-015-9519-4), perhaps because “many of the early promises of immunoassay did not fully materialize, as shown by reports of extensive purification being important prior to assay”.  The method used here may be okay for bacterial cultures, but is dubious for plant extracts as used.  This is moreover a quandary as a GC-3Q-MS was employed for a much more mundane role in amino acid analysis.  Exacting measurements of IAA levels in specific tissues and of auxin responsive gene expression is necessary if the authors which to make the point they express in the title.   

Author Response

The basic premise of this manuscript is that rhizobacteria that overproduce indole-3-acetic acid have the capacity to increase metabolic activity resulting in biomass increase.   As the authors note, this has been looked at previously by a number of different approaches and with different results, leading to ambiguous interpretations.  Resolution of these issues would be important.

This present manuscript is difficult for reviewers because of the incessant use of… “according to manufacturer’s instructions” and “determined as described by…” in the place of actually telling the reader what they did.  This can be anything from just annoying to highly irritating (like the Lichtenthaler reference where a huge chapter is cited, but I assume they only needed to point to Table III – or better they could have said succinctly what they did then give the citation! Also, the Bucciarelli et al paper that you have to look up to learn “chemically scarified with concentrated sulfuric acid for 8 min and surface sterilized for 3 min with commercial-grade bleach”).  Of course, an important reviewer task is to ask if the methods were suitable, so tracking down a dozen or so citations is a pain especially given that our time is a free gift to the authors and the journal.

Reply – We have considered your comments and introduced additional details in the Methods section. The wording “according to the manufacturer’s instructions” was left only when the protocol used was exactly the one reported in the manufacturer’s manual and when this manual was easily available on the web.

Moving from that aspect to a more serious issue is the methods section on “Determination of free amino acids, soluble sugars and organic acids” where the section is almost entirely devoted to amino acid analysis.  The procedure used is not the best (Artifacts in Trimethylsilyl Derivatization Reactions and Ways to Avoid Them, https://littlemsandsailing.files.wordpress.com/2014/04/silyl_2014_text.pdf), but is acceptable for amino acids.  Soluble sugar analysis is barely described in this present manuscript and typically TMS reagents react poorly to sugars, often giving multiple peaks and missing sugars in complex plant samples (Zarate et al. 2017 doi:10.3390/metabo7010001). 

Reply – We agree that the procedure used in our metabolic analysis is more appropriate for amino acids than for sugars. But our aim was not to analyse all the sugars present in the extracts. During the amino acids analysis we noticed that the peaks of these sugars were particularly abundant and therefore we focused our attention on these peaks and identified them. To obtain the relative abundances of the corresponding metabolites, the peak areas of different TMS derivatives of the sucrose, fructose and glucose were summed and divided by the peak area of internal standard.

The authors state “To analyse the effect of IAA-overproduction inside RD64 bacteroids on the expression of genes involved in nitrogen fixation, qRT-PCR analysis of selected genes was carried out on root nodules of Medicago plants infected with the strains E. meliloti RD64 and 1021 at different DAI.”  There is a curious omission here in that they examined ‘selected genes’ important for nitrogen metabolism, but omitted genes that would have confirmed or negated their central hypothesis regarding IAA “overproduction” – namely they did not look for elevation in the   IAA, GH3, and SAUR families, as well as several PIN genes (Paponov et al. 2008 https://doi.org/10.1093/mp/ssm021).  Such analysis might have helped to understand if indeed additional IAA production is being perceived by the plant auxin signaling systems.  Without this information, the association of the general metabolic responses noted with elevated auxin is mostly just correlation and a bit of conjecture.

Reply – Further to your suggestion further qRT-PCR experiments have been carried out results and the results concerning the expression levels of eleven other genes have been included in the revised manuscript. The new selected genes belong to the following functional classes: metabolism, environmental information processing, and genetic information processing. One gene belongs to the major auxin early response gene families. The overall data clearly show that: 1) IAA influences (induces or reduces) the expression of different genes, including the ones involved in nitrogen-fixation; 2) the IAA-overproduction inside RD64 bacteroids is perceived by the plant auxin signalling systems (Results section and revised Figure 1).

They have a problem with history.  They state “The main plant auxin, IAA, was identified in 1937” but from plants this is usually attributed to Berger and Avery in 1944, although it was identified from urine and called ‘heteroauxin’ a decade earlier.  Likewise, many of the earlier studies of indole metabolism altered ‘brown rhizobia’ and of IAA involvement go uncited (see, for example, Williams and Signer 1990). 

Reply – We agree with your comment and introduced the reference of “Berger and Avery” in the revised manuscript.

They also note “…plant hormone biotechnology has been to modulate their internal concentration by modification of either synthesis or degradation pathways, or both [2].” They seem to ignore the classic work of Glass and Kosuge on the  indole-3-acetic acid-lysine synthetase (iaaL gene) (Spaepen and Vanderleyden, 2010  doi: 10.1101/cshperspect.a001438) that is a very effective method for alteration of IAA levels in planta (Romano et al. 1991 DOI: 10.1101/gad.5.3.438). 

Reply – Thanks to your comment we have realized that the concept concerning the modulation of hormone levels and metabolism to improve plant growth in adverse conditions as written led to wrong interpretation. Therefore, we introduced changes in the text of the Introduction section.

Finally, and importantly, a manuscript on the effect of indole-3-acetic acid on plant yield needs to show with some chemical certainty that IAA levels are indeed elevated.  Very few labs use immunoassays for IAA levels in plants, as noted in their review (Tivendale and Cohen 2015 DOI: 10.1007/s00344-015-9519-4), perhaps because “many of the early promises of immunoassay did not fully materialize, as shown by reports of extensive purification being important prior to assay”.  The method used here may be okay for bacterial cultures, but is dubious for plant extracts as used.  This is moreover a quandary as a GC-3Q-MS was employed for a much more mundane role in amino acid analysis.  Exacting measurements of IAA levels in specific tissues and of auxin responsive gene expression is necessary if the authors which to make the point they express in the title.   

Reply – It has been previously demonstrated by TMS GC-MS analysis that the expression of the chimeric operon p-iaaMtms2 in E. meliloti bacteroids (strain RD64) resulted in a 10-fold increase of IAA content in the root-nodules of M. sativa: the IAA concentration measured for nodules of Ms-RD64 was 0.12 nmol /g FW, whereas the one estimated for nodules of Ms-1021 was 1.2 nmol /g FW [49]. When the nodulated plants were analyzed by confocal microscopy, using an anti-IAA antibody, a stronger signal was observed in bacteroids of RD64-nodulated plants, even when they were located in the senescent nodule zone [25]. The substantial increase in the expression level of GH3, which belongs to the major auxin early response gene families, here reported confirmed those results. The already known data concerning the IAA levels in nodules of plants infected with RD64 and 1021 has now been introduced in the manuscript (Results section).

Round 2

Reviewer 2 Report

I still do not fully understand how the sugars were measured.  They say “To obtain the relative abundances of the corresponding metabolites, the peak areas of different TMS derivatives of the sucrose, fructose and glucose were summed and divided by the peak area of internal standard.” However, we are not given any sugar IS, so are they comparing with norleucine?  Is it being assumed that the detector response for the multiple peaks that are summed are the same?  Nothing is said about retention time matching and how it was done, as the MS spectra for hexoses, for example, are very similar under these conditions.  As they note in their response, the analysis of sugars was because they noticed them.  This is not a very good reason to not follow through with a more complete study – if the three most abundant are interesting, why wouldn’t the less abundant ones also be interesting? Similarly for the organic acids…I assume that this was also the ‘big peak’ approach?  At least be clear to the reader that the methods were designed to NOT be comprehensive – but this is kind of unsatisfying isn’t it?   

[Line 14 page 10] They describe malate and citrate as being “derived through sucrose metabolism”.  This is not exactly correct as they are better described as TCA cycle intermediates.  Sucrose must go through invertase and glycolysis before entering into the the TCA cycle, so although sucrose can be a source for these acids, it is not the obligate source.  

I also remain puzzled by the use of ELISA here for IAA measurements. They now cite the Pii et al [49] paper that shows IAA by TMS GC-MS.  The values reported were 0 – 1.2 nmol/g FW for total (free IAA plus IAA released by base hydrolysis).   They say their analysis “revealed that the leaves of 28-day-old Ms-1021 and Ms-RD64 plants had almost the same IAA content (Table 1).”  But they do not discuss the fact of Pii showing peak levels of 1.2 nmol/g FW and Table 1 shows a leaf content of 60 µmol/g FW [60,000 nmol/g FW], a quite remarkable difference and this level of IAA, over 10 mg/ml, is probably not soluble in water.  In my last review, I cited a review article on reliability of ELISA, but that did not seem to inspire the authors much.  I think the original reference on comparison of ELISA assay for IAA with analysis by gas chromatography selected ion monitoring mass spectrometry deserves some reflection (see DOI:  10.1104/pp.84.4.982) as that paper says you need rather exhaustive purification (not employed in this present manuscript work) to get even close to useful data.  Basically, the IAA data in this paper is difficult to believe and the methods quite suspect.    

“We agree with your comment and introduced the reference of “Berger and Avery” in the revised manuscript.” – thank you, but in your reference list [1] in the citation you sadly left poor Professor Berger out.

The addition of the GH3 analysis is helpful, but, of course not clear-cut, as GH3 family is responsible for both IAA and JA conjugation, so this one gene family is not likely a complete story.  The increase in GH3 is important but is not as strong as some of the other changes noted, suggesting either it is a poor report of that levels of IAA are not changing enough to trigger a bigger response.